# High-throughput screening for discovery of benchtop separations systems for selected rare earth elements

Joshua J.M. Nelson [1], Thibault Cheisson [1,2], Haley J. Rugh [1], Michael R. Gau[1], Patrick J. Carroll[1] &
Eric J. Schelter [1]*

Rare earth (RE) elements (scandium, yttrium, and the lanthanides) are critical for their role in sustainable energy technologies. Problems with their supply chain have motivated research to improve separations methods to recycle these elements from end of life technology. Toward this goal, we report the synthesis and characterization of the ligand tris[(1-hydroxy-2-oxo-1, 2-dihydropyridine-3-carboxamido)ethyl]amine, $H_3$1·TFA (TFA = trifluoroacetic acid), and complexes 1·RE (RE = La, Nd, Dy). A high-throughput experimentation (HTE) screen was developed to quantitatively determine the precipitation of 1·RE as a function of pH as well as equivalents of $H_3$1·TFA. This method rapidly determines optimal conditions for the separation of RE mixtures, while minimizing materials consumption. The HTE-predicted conditions are used to achieve the lab-scale separation of Nd/Dy ($SF_{Nd/Dy}$ = 213 ± 34) and La/Nd ($SF_{La/Nd}$ = 16.2 ± 0.2) mixtures in acidic aqueous media.

---

[1] P. Roy and Diana T. Vagelos Laboratories, Department of Chemistry, University of Pennsylvania, 231S. 34th Street, Philadelphia, PA 19104, USA. [2] Present address: Eramet Ideas, 1 avenue Albert Einstein, 78190 Trappes, France. *email: schelter@sas.upenn.edu

Clean energy technology is increasingly reliant on rare earth elements (RE: Sc, Y, and La–Lu). For example, rare earth (RE) elements are critical components of hybrid car batteries, lighting phosphors, and permanent magnets[1–3]. For these applications, precise blends of individual REs are often required. In recent years, volatility in the global RE market[4–6] has directed attention toward new methods to recycle REs from waste electronic and electrical equipment (WEEE), since recycling represents only a small fraction of the supply chain for these elements[2,7–9].

RE separations are performed industrially using countercurrent solvent extraction. This method requires large volumes of solvents due to relatively dilute operating conditions (20 ppm ~ 0.1 M RE)[10–12]. Opportunities remain to improve selectivity for individual rare earths over a single extraction and stripping step. Toward these goals, researchers have developed novel ligands[13], ionic liquids[14,15], and extractants[16–18]. These findings are potentially compatible with existing countercurrent solvent extraction technology. However, the investment required for countercurrent extraction is a barrier to recycling REs from WEEE, and is only economically viable if the price of rare earth oxides remains high[19]. There is thus a clear need for alternative RE recycling methods to countercurrent solvent extraction. Selective precipitation of individual REs from mixtures, isolated with a simple filtration step, could meet this need. Such goals are especially pertinent to the binary mixtures of rare earths used in technology. Mixtures of La/Nd are present in nickel metal hydride batteries, and Nd/Dy mixtures are used in permanent magnets[7]. Recent advances in separating RE mixtures have been accomplished through photochemical reduction[20], selective crystallization[21,22], chromatographic separations[23,24], and the use of supported liquid membranes[25,26].

High-throughput experimentation (HTE) refers to running multiple reactions in parallel[27]. Such methods have been used extensively in catalysis to rapidly screen reaction conditions (e.g., metal ion, ligand, solvent) and require a fraction of the time and materials resources necessary for lab-scale methods[28–30]. HTE methods have been used to assess drug and protein solubility in aqueous media[31,32], and to determine conditions for precipitation or protein crystallization[33,34]. However, HTE methods have not been used to screen RE separations conditions.

Our group previously reported a chelating tris-hydroxylamine proligand tris(2-tert-butylhydroxylamine)benzylamine (H$_3$TriNOx), which demonstrated high separation factors over a single leaching step for pairs of REs ($SF_{Nd/Dy}$ ~300, $SF_{La/Nd}$ ~10)[35–38]. From a practical standpoint, the hydroxylamine moieties required the use of strong bases—incompatible with water—to coordinate the RE ions. Another issue was the human and environmental toxicity of the organic solvents used (benzene, toluene, or n-hexane) in that system. To address these limitations, we set out to develop a water-soluble and -stable ligand framework that would deliver comparable separations performance to H$_3$TriNOx. Taking inspiration from the Raymond group's use of the hydroxypyridone motif (HOPO, Fig. 1) due to its high affinity for REs in aqueous conditions[39–42] and potential biomedical applications[43–46], we synthesized the novel proligand tris[(1-hydroxy-2-oxo-1,2-dihydropyridine-3-carboxamido)-ethyl]amine, H$_3$tren-1,2,3-HOPO·TFA (H$_3$1·TFA). We

hypothesized that exchanging the positions of the hydroxyl and carbonyl moieties would maintain the high affinity of HOPO-derivates for REs while allowing for bridging interactions that were critical for the separations selectivity of our H$_3$TriNOx system[35–38].

We herein describe the synthesis and characterization of H$_3$1·TFA, its related RE complexes (RE = La, Nd, Dy), and its application in the separation of binary mixtures of REs via selective precipitation from aqueous media. To aid in this effort, we have developed a HTE screen to optimize precipitation-based RE separations.

## Results

**Synthesis of ligand and complexes**. The new proligand H$_3$tren-1,2,3-HOPO (H$_3$1·TFA), was synthesized in good yield (56%) from 1-hydroxy-2-oxo-1,2-dihydropyridine-3-carboxylic acid and tris(2-aminoethyl)amine (tren) and isolated as a trifluoroacetic acid (TFA) salt (Fig. 2, see Supplementary Methods for full synthetic details). Importantly, H$_3$1·TFA was synthesized without any protection/deprotection steps commonly used in the synthesis of HOPO-based ligands, which improved atom economy and minimized the total number of steps required[47]. The ligand H$_3$1·TFA has limited stability in saturated aqueous Na$_2$CO$_3$ solution (<24 h), but is stable for >4 months as a solution in 2 M HCl, and indefinitely as a solid.

The complexes, 1·RE (RE = La, Nd, Dy), were synthesized by stirring a solution of RECl$_3$·nH$_2$O with one equivalent of H$_3$1·TFA in H$_2$O for 3 h without addition of base. The $^1$H NMR spectra in d$_6$-DMSO demonstrated the complexes to be $C_{3v}$-symmetric in solution. X-ray diffraction analysis of crystals grown by vapor diffusion of H$_2$O into concentrated dimethylformamide (DMF) solutions revealed nearly isostructural motifs for the three RE complexes. The lanthanide cations were 8-coordinate and bound to all 6 oxygen atoms of the hydroxypyridone rings. The coordination sphere included an apical DMF molecule and an equatorial water molecule (Fig. 3). The crystal packing revealed the formation of 1-D chains through the water molecules H-bonding network (Supplementary Fig. 3).

While the solid-state speciation of all 1·RE complexes was similar, we noticed a marked dependence on the solution pH for the precipitations of the individual complexes, which may be due to differences in the pK$_a$ values of the ligand among the RE complexes. For example, precipitation from 1 M HCl could be achieved for RE = Nd, Dy, but not La. This prompted investigation into use of H$_3$1·TFA as a material for the separation of RE mixtures in acidic media.

**High-throughput experimentation**. HTE methods allow for the rapid screening of multiple reaction variables while minimizing

**Fig. 1 Ligand frameworks.** H$_3$TriNOx ligand developed by the Schelter group[36], HOPO-based ligand developed by the Raymond group[40], and ligand developed in this work.

**Fig. 2 Preparation of H$_3$1·TFA and 1·RE.** Synthesis of H$_3$1·TFA from 1-hydroxy-2-oxo-1,2-dihydropyridine-3-carboxylic acid and formation of **1·RE** (RE = La, Nd, Dy).

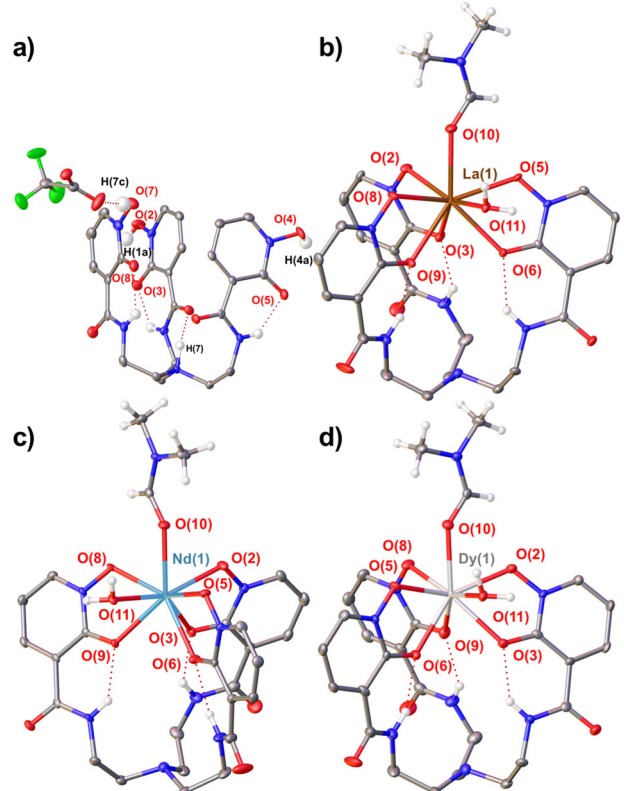

**Fig. 3 Solid state structures of H₃1·TFA and 1·RE.** Thermal ellipsoid plots of H$_3$1·TFA (**a**), **1·La**(DMF)(H$_2$O) (**b**), **1·Nd**(DMF)(H$_2$O) (**c**), and **1·Dy** (DMF)(H$_2$O) (**d**).

resource consumption. Here, we were interested in developing an assay to quantify the precipitation of **1·RE** as a function of pH. We were particularly interested in determining the conditions with the largest difference in precipitation among REs. These conditions would theoretically yield the greatest separation of RE mixtures.

Precipitation experiments were performed in 96-well plates at 0.10, 0.25, 0.50, 1.00, 1.50, 2.00 M HCl with 1.0, 1.5, and 2.0 equivalents of H$_3$1·TFA to RECl$_3$ (RE = La, Nd, Dy) (Fig. 4). The effect of chloride concentration was also investigated using 1.0 equivalent of H$_3$1·TFA with 1.00 M KCl$_{(aq)}$ as a solvent (see Supplementary Methods for details). Combined, this provided 19 different reaction conditions for each RE. The reaction mixtures were agitated in well plates for 24 h, then filtered in parallel using filter plates. The filtrates were analyzed for RE-content using inductively-coupled plasma optical emission spectroscopy (ICP-OES), and yields calculated from these values (see Supplementary Eq. 1 for yield calculation). The results of the assay are summarized in Fig. 4c.

The amount of RE precipitated decreased with increasing HCl concentration from 0.10 M to 2.00 M. Using 1.00 M KCl$_{(aq)}$ as solvent resulted in similar precipitation results as 0.10 M HCl. These results suggested that precipitation is inhibited by increasing proton concentration in solution, and that the chloride anion does not significantly inhibit **1·RE** formation and precipitation. Increased yields of extracted RE in the solid portion were observed with increasing equivalents of H$_3$1·TFA.

Surprising here was the notable difference in precipitation behavior among the three RE complexes. **1·La** ceased to precipitate from solution starting at 0.25 M HCl, as evidenced by a yield ~0%, where **1·Nd** and **1·Dy** continued to precipitate from solution through 1.00 M HCl and 2.00 M HCl, respectively. These data suggested it should be possible to selectively

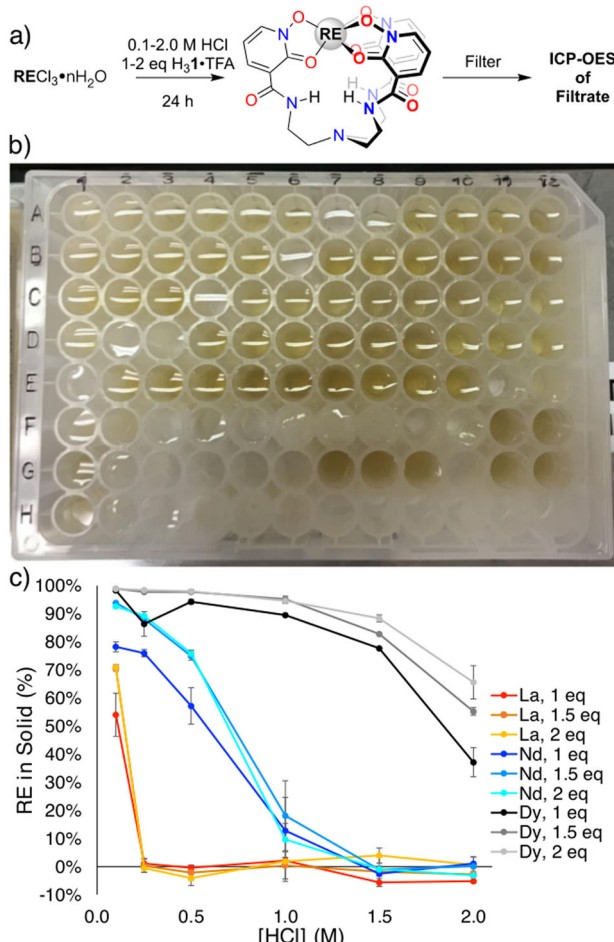

**Fig. 4 HTE screening. a** General schematic for the HTE precipitation screen. **b** Image of the precipitation screen. **c** Results from the HTE precipitation screen. Error bars represent the standard deviation between three trials.

precipitate a single RE from a mixture of REs by controlling the pH, through complexation with the ligand.

**Separations experiments based on HTE.** According to the HTE data, the largest difference in RE precipitation for Nd and Dy occurred at 1.50 M HCl using 2.0 equivalents H$_3$1·TFA. To validate these conditions for lab scale separations experiments, we first examined the separation of 1:1 Nd/Dy mixtures using one equivalent of H$_3$1·TFA. Allowing the reaction mixture to stir in neutral water for 1 h prior to filtration through a fine porosity sintered glass frit resulted in $SF_{Nd/Dy} = 28.1 \pm 0.8$ (Table 1, Entry 1). Repeating the experiment with 1.0 M HCl as the solvent resulted in $SF_{Nd/Dy} = 28.9 \pm 3.2$ (Table 1, Entry 2), which is consistent within error. However, the enrichment factor of the solid portion for 1.0 M HCl is double that of neutral water as solvent. This result suggested that less **1·Nd** precipitated from solution—as predicted by the HTE data—resulting in a more pure solid phase. Using two equivalents of H$_3$1·TFA improved the separations achieved in 1.0 M HCl to $SF_{Nd/Dy} = 46.9 \pm 4.9$ (Table 1, Entry 3). Two equivalents of H$_3$1·TFA in 1.5 M HCl further improved the separation factor from 1:1 Nd/Dy mixtures to $SF_{Nd/Dy} = 71.4 \pm 8.0$ (Table 1, Entry 4). Together, these preparatory-scale results validated the HTE-predicted optimal conditions for separations of 1:1 mixtures of Nd/Dy.

To further optimize the Nd/Dy separations, we investigated the effect of reaction time and metal concentration. Extending the

**Table 1 Optimization of rare earth separations conditions using H₃1·TFA.**

| | | | | ICP-OES results[a] | | | Avg. % distribution/purity | |
|---|---|---|---|---|---|---|---|---|
| Entry | RE1:RE2:H₃1 | Solvent | Time (h) | $EF_{solid}$ | $EF_{filtrate}$ | $SF_{RE2/RE1}$ | Solid (% RE1) | Filtrate (% RE2) |
| 1 | 1 Dy:1 Nd:1 H₃1 | H₂O | 1 | 7.47 ± 0.23 | 3.76 ± 0.23 | 28.1 ± 0.8 | 88.2 ± 0.3 | 79.0 ± 1.0 |
| 2 | 1 Dy:1 Nd:1 H₃1 | 1.0 M HCl | 1 | 15.5 ± 0.9 | 1.86 ± 0.11 | 28.9 ± 3.2 | 93.9 ± 0.3 | 65.0 ± 1.3 |
| 3 | 1 Dy: 1 Nd:2 H₃1 | 1.0 M HCl | 1 | 6.09 ± 0.26 | 7.70 ± 1.13 | 46.9 ± 4.9 | 85.9 ± 0.5 | 88.4 ± 1.5 |
| 4 | 1 Dy:1 Nd:2 H₃1 | 1.5 M HCl | 1 | 23.8 ± 1.5 | 3.00 ± 0.26 | 71.4 ± 8.0 | 96.0 ± 0.2 | 74.9 ± 1.6 |
| 5 | 1 Dy:1 Nd:2 H₃1 | 1.5 M HCl | 3 | 31.0 ± 1.2 | 3.24 ± 0.09 | 100 ± 6 | 96.9 ± 0.1 | 76.4 ±± 0.5 |
| 6[b] | 1 Dy:1 Nd:2 H₃1 | 1.5 M HCl | 3 | 15.6 ± 2.1 | 8.93 ± 2.00 | 138 ± 29 | 93.9 ± 0.7 | 89.7 ± 1.9 |
| 7 | 1 Dy:1 Nd:2 H₃1 | 1.5 M HCl | 24 | 31.6 ± 1.7 | 4.07 ± 0.87 | 128 ± 21 | 96.9 ± 0.2 | 79.9 ± 3.5 |
| 8[b] | 1 Dy:1 Nd:2 H₃1 | 1.5 M HCl | 24 | 12.1 ± 1.8 | 17.6 ± 0.2 | 213 ± 34 | 92.3 ± 1.1 | 94.6 ± 0.1 |
| 9[b,c] | 1 Dy:19 Nd:2 H₃1 | 1.25 M HCl | 24 | 0.33 ± 0.05 | 21.1 ± 2.4 | 6.88 ± 0.36 | 24.7 ± 2.9 | 95.4 ± 0.5 |
| 10[b] | 1 Nd:1 La:2 H₃1 | 0.25 M HCl | 24 | 1.57 ± 0.11 | 10.4 ± 0.7 | 16.2 ± 0.2 | 61.0 ± 1.6 | 91.2 ± 0.6 |

Reactions were performed at 19.4 mM RE1 under ambient conditions unless otherwise specified. Errors are reported as the standard deviation between three trials
[a]$EF$ enrichment factor; $SF$ separations factor; see Supplementary Eqs. 2–5 for calculations used to determine these values
[b]Performed at 39.5 mM RE1
[c]Adjusted from 1.5 M HCl using 1.0 M NaOH

reaction time to 3 or 24 h resulted in $SF_{Nd/Dy} = 100 ± 6$ and $128 ± 21$, respectively (Table 1, Entries 5, 7). The 3 h experiment was repeated at double the RE concentration in solution—achieved by eliminating half of the volume of acid—resulting in $SF_{Nd/Dy} = 138 ± 29$ (Table 1, Entry 6). The 24 h and concentrated 3 h separations results were identical, within error. While the separation factors were equivalent, it is worth noting that the enrichment factors of the solid ($EF_s$) was halved and the enrichment factor of the filtrate ($EF_f$) was doubled for the shorter, more concentrated experiment as compared to the 24 h experiment. Increasing the concentration and allowing the experiment to run for 24 h resulted in the highest separations factor achieved, $SF_{Nd/Dy} = 213 ± 34$ (Table 1, Entry 8). From these experiments, 78% of 1·Dy was recovered in the solid portion, and 88% of the Nd-enriched filtrate could be recovered, revealing there to be minimal loss of material during the filtration process. This result demonstrated the achievement of an effective, efficient separation of neodymium and dysprosium with minimal quantities of dilute acid followed by only a quick rinse using minimal quantities of water. By comparison, the commercially relevant 2-ethylhexyl-mono(2-ethylhexyl) ester phosphonic acid (HEHEHP) and CYANEX® 572 deliver calculated separations of $SF_{Nd/Dy} = 50$ and $69.5$, respectively (Table 2)[48]. Under the conditions reported, these extractants require more than double the volume of solvent, as well as an organic phase to achieve $SF_{Nd/Dy}$ values less than one third of that achieved using H₃1·TFA. However, it is worth noting that performance of phosphorous-based extractants is highly dependent on the diluents used, with some reports achieving comparable separations to H₃1·TFA[10–12].

To test the utility of these conditions in application to relevant mixtures in electronic waste, we made a 5% Dy mixture in Nd, and added 2 equivalents of H₃1·TFA per Dy. Surprisingly, no precipitation was observed within 24 h. Slowly adjusting the acid concentration with 1.0 M NaOH to ~1.25 M HCl eventually resulted in the formation of a precipitate. The solid was enriched to 24.7 ± 2.9% in dysprosium—a fivefold increase from the starting mixture in one step (Table 1, Entry 9). The filtrate did not exhibit significant enrichment in neodymium.

The optimized Nd/Dy separation conditions were applied to La/Nd mixtures. The largest difference in precipitation from the high-throughput screen was achieved using 0.25 M HCl as the solvent and 2 equivalents H₃1·TFA. This resulted in $SF_{La/Nd} = 16.2 ± 0.2$ (Table 1, Entry 10). This value is greater than the separations achieved by HEHEHP and CYANEX® 572, $SF_{La/Nd} = 9.0$ and $14.6$, respectively, require the use of an organic diluent, and greater quantities of extractant (Table 2). Using H₃1·TFA, the enrichment of the solid was low, $EF_s = 1.57 ± 0.11$, while the enrichment of the filtrate was much greater, $EF_f = 10.4 ± 0.7$ (Table 1, Entry 10). Evidently, a significant portion of lanthanum precipitated, which lowered $EF_s$ and resulted in a good $EF_f$. Optimization of reaction times and metal concentrations will further improve the performance of H₃1·TFA for the separation of La/Nd mixtures.

**Ligand recovery**. Mirroring ligand-stripping practices commonly employed in solvent extraction, we were interested in recovering H₃1 from the purified RE salts for reuse in additional separations. We found that ~20 mg 1·Dy could be dissolved in 0.4 mL 12 M HCl, presumably forming H₃1·HCl and DyCl₃·nH₂O in solution. Addition of 2.0 mL EtOH resulted in the formation of a precipitate, which was determined to be H₃1 with a residual ~12% 1·Dy by [1]H NMR analysis (see Supplementary Methods for experimental details). Considering the ligand could be reused for additional Nd/Dy separations, this minor impurity does not pose an operational issue. This stripping step was able to recover 84% of the ligand and 86% DyCl₃ while using minimal solvent volumes.

## Discussion

H₃1·TFA and its related complexes, 1·RE (RE = La, Nd, Dy), were synthesized in good yields and characterized in solution by [1]H NMR spectroscopy and in the solid state using single crystal X-ray analysis. H₃1·TFA exhibits high stability under acidic conditions required for solubilizing RE oxides. We have demonstrated the effect of acid concentration and equivalents of H₃1·TFA on the precipitation of 1·RE from solution using HTE screening. From this screen, we found that La and Nd did not precipitate from solutions at 0.25 M and 1.50 M HCl, respectively,

**Table 2 Comparison of HEHEHP and CYANEX® 572 with H$_3$1·TFA.**

| Metric | HEHEHP | CYANEX® 572 | H$_3$1·TFA |
|---|---|---|---|
| Reference | 48 | 48 | This work |
| Solvent | HCl/organic diluent | HCl/organic diluent | HCl |
| Solvent Hazards | corrosive/flammable, harmful to environment | corrosive/flammable, harmful to environment | corrosive |
| [RE1] (mM) | 20 | 20 | 39.5 |
| Equivalents Extractant | 10 | 10 | 2 |
| $SF_{La/Nd}$ | 9.0 (10)[a] | 14.6 | 16.2 |
| $SF_{Nd/Dy}$ | 50 (200)[a] | 69.5 | 213 |

[a]Approximate SF achieved using 20 ppm RE in 2% nitric acid, 0.5 M extractant in dodecane[10]

whereas Dy precipitated from up to 2.00 M HCl solutions. These results provided optimal reaction conditions for the separation of 1:1 RE mixtures in a single complexation/separation step. This resulted in $SF_{Nd/Dy} = 213 \pm 34$ and $SF_{La/Nd} = 16.2 \pm 0.2$. This method was also applied to 5% Dy in Nd mixtures, and enriched the solid to ~25% Dy in a single step. The ligand could be recovered in 84% yield with only minor residual **1·Dy** under mild conditions. Our system was found to be comparable to and in some cases outperform currently relevant industrial counter-current solvent extractants HEHEHP and CYANEX® 572 in terms of separations achieved in a single step, and total solvent usage. The origin of the selectivity of this system is potentially due to differences in the pK$_a$ values of the ligand among the different REs, which results in the formation of species with differing solubility at varying pHs. Preliminary results suggest that the filtrate from the optimized Nd/Dy separations mixture comprised a more complicated speciation than simple chloride salts; evident in the infrared spectra of the solid and filtrate portions obtained from the separations experiments (Supplementary Figs. 14 and 15). Expanded work to identify the origin of selectivity of this system are on-going.

The HTE methodology can easily be applied to other water-soluble ligands to rapidly screen conditions for the separation of RE mixtures creating minimal waste, an area of ongoing interest in our group.

## Methods

**High throughput precipitation screening**. High throughput precipitation screening experiments were performed in 96-well reaction plates with a maximum volume of 2.00 mL per well. Experimental wells (EW) were loaded with 250 μL of the appropriate individual RECl$_3$ (66 mM, RE = La, Nd, Dy) solution followed by 600 μL of the appropriate ligand solution for a total volume of 850 μL ([RE]$_{initial}$ = 19.4 mM). Each set of experimental conditions were replicated in triplicate. Positive control (PC) wells were loaded with 250 μL of all three RECl$_3$ (RE = La, Nd, Dy) solutions and 100 μL solvent for a total volume of 850 μL and were placed after every nine experimental wells. The well plate was covered with an adhesive aluminum foil cover to prevent solvent evaporation and cross-contamination between wells, then placed on an innova2180 platform shaker moving at 330 RPM for 24 hours. The reaction plate was centrifuged at 2000 RPM on a GeneVac EZ-2 Personal Evaporator for 1 h. The supernatant was transferred to a 96-well filter plate with a maximum volume of 1.00 mL per well. Dynamic vacuum was applied, and the filtrate collected in a 96-well collection plate with a maximum volume of 2.00 mL per well. The filtrate was analyzed for metal content by ICP-OES. See Supplementary Methods and Supplementary Table 1 for ICP-OES details. See Supplementary Eq. 1 for yield calculation.

**General procedure for separation of rare earth mixtures**. To a stirring solution of H$_3$**1**·TFA (1–2 equivalents) in the appropriate solvent (3.00 mL) was added a solution of **RE1**Cl$_3$·nH$_2$O (0.083 mmol) and **RE2**Cl$_3$·nH$_2$O (0.083 mmol) in the same solvent (1.25 mL). At the end of the reaction time (1–24 h), the mixture was filtered through a fine porosity sintered glass frit, and the solid washed twice with H$_2$O (0.5–1.0 mL), and once with acetone (0.5 mL, optional wash). The solid was dried on the frit, and the filtrate evaporated. The RE content of the solid and filtrate portions was analyzed by ICP-OES. All separation experiments were performed in triplicate. See ESI for calculation of enrichment and separations factors.

## Data availability

Data to support the conclusions in this paper are available in the main text or the supplementary materials (NMR spectra, Supplemental Figs. 2–13; FT-IR spectra, Supplemental Figs. 14 and 15) and are available from the authors upon reasonable request. The X-ray crystallographic coordinates for structures reported in this study have been deposited at the Cambridge Crystallographic Data Centre (CCDC), under deposition numbers 1906140-1906143. These data can be obtained free of charge from The Cambridge Crystallographic Data Centre via http://www.ccdc.cam.ac.uk/data_request/cif.

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

## Acknowledgements

The authors gratefully acknowledge support from the University of Pennsylvania and the U.S. Department of Energy, Office of Science, Office of Basic Energy Sciences, Separation Science program under Award DE-SC0017259. We acknowledge the Center for Actinide Science and Technology (CAST), an Energy Frontier Research Center (EFRC) funded by the U.S. Department of Energy, Office of Basic Energy Sciences (DE-SC0016568) for support of T.C.

## Author contributions

J.J.M.N., T.C. and E.J.S. conceived the project and designed the experiments. J.J.M.N. and H.J.R. performed the experiments and analyzed the data. M.R.G. and P.J.C. performed X-ray crystallography. J.J.M.N., T.C. and E.J.S. wrote the manuscript. All authors discussed the results and approved of the manuscript.

## Competing interests

There authors declare no competing interests.
