## [Peer Review File · Communications Chemistry]

Reviewers' comments:

Reviewer #1 (Remarks to the Author):

The manuscript entitled as "High-Throughput Screening for Discovery of Benchtop Separations Systems for Selected Rare Earth Elements" is demonstrating that the new ligand H31·TFA system, stemming from two reported ligands with their respective high affinity or separation selectivity towards REs, shows high separation factors over a single step for pairs of REs (SFNd/Dy=213±34 and SFLa/Nd=16.2±0.2). Inspired by the observation of different precipitation behavior of 1·RE complexes under the same pH condition, the effects of acid concentration and equivalents of H31·TFA on the precipitation of corresponding RE complexes were further studied in a systematic way, setting up an efficient screening method for predicting the optimal separation condition for binary mixtures of rare earth. It was verified by the subsequent detailed experiments, also taking consideration of reaction time and metal concentration. Compared with the current industrial means for the separation of the identical pair of REs by using two commercial extractants, this method herein introduced features the less time- and (harmful) solvent- consumption, higher purity of separated REs and recyclability of the materials. Although the reason for the selectivity of this system is still indefinite, this work indeed showcases a promising separation application from a practical standpoint. Overall, the manuscript is suitable for the Communications Chemistry and I recommend acceptance after minor changes.

1. Ligand H31 was described to exist as the trifluoroacetic acid salt (H31·TFA), mainly deduced from the obtained crystal structure of it.
 - a. The crystallization detail of H31·TFA should be reported in ESI. Besides, different crystallization conditions can affect the final crystal structure of product. Is it possible that the obtained product co-crystallizes with TFA, thus yielding the trifluoroacetic acid salt?
 - b. How to determine the hydrogen atom on the protonated bridging N atom? Is there reasonable electron density to confirm the existence of this hydrogen atom?
 - c. The direct evidence for the pure form of H31·TFA should be the NMR spectroscopy with the internal standard, as the author mentioned. The corresponding spectra should be added in the ESI.
 - d. Maybe element analysis will be helpful for further confirmation of the exact form.
2. As described in the header of Table 1, there are many factors corresponding to different properties of RE1 and(or) RE2. Among them, the SFRE1/RE2 should be corrected as SFRE2/RE1, then it can match the mentioned SFNd/Dy in the main text.
3. Two IR spectra are added at the end of ESI. There is no information/explanation about these two spectra in the main text and ESI. Adding one sentence in the figure caption to interpret may be better for the reader to understand.

Reviewer #2 (Remarks to the Author):

Please see attached

Reviewer #3 (Remarks to the Author):

The manuscript contains some interesting experimental findings on the selective precipitation of Ln (La, Nd and Dy) with a newly synthesized ligand. However, the paper aims at developing a technology for Ln recycling, but the proposed approach is far from being superior to the current technologies.

First- the authors claim that "... solvent extraction can require hundreds of hours to achieve purified REs, compared to the less than 48 hours required for our system including ligand recovery

steps (vide infra).¹⁶ Reference 16 is hardly relevant here as it describes the application of Aliquat 336 in a certain system. While the authors compare their method with HEHEHP and Cyanex 572 in terms of separation factors, they do not cite any literature describing the corresponding processes (Ref 46 does not point to any process conditions). As these processes are counter-current, very high overall separation factors (1,000-10,000) can be accomplished within 10-20 stages, and the duration of the process should not exceed several hours even if the slowest equipment (e.g. mixer-settler) is used.

Second- The SF for La/Nd for the proposed method is only 16, and it would take 3-4 precipitation steps (6-8 days) to get to a decent overall separation.

Third issue is the Ln recovery and the recycling of the chemicals: according to the paper: " ^{147}Dy could be dissolved in 0.4 mL 12 M HCl, presumably forming $\text{H}_3^{147}\text{DyCl}_3$ and $\text{DyCl}_3 \cdot n\text{H}_2\text{O}$ in solution. Addition of 2.0 mL EtOH resulted in the formation of a precipitate, which was determined to be $\text{H}_3^{147}\text{Dy}$ with a residual $\sim 12\%$ ^{147}Dy by ^1H NMR analysis (see ESI for experimental details). Considering the ligand could be reused for additional Nd/Dy separations, this minor impurity does not pose an operational issue. This stripping step was able to recover 84% of the ligand and 86% DyCl_3 under relatively mild conditions while using minimal solvent volumes." It is unclear why using concentrated HCl is called "mild conditions". Using ethanol to precipitate the ligand, followed by the filtration of the solid and its dissolution will further increase processing time. Also, returning 12% Dy to the separation step will eventually saturate the system with Dy and will lower the purity of Nd.

The manuscript should not be published in this current form; it should be submitted to a specialized inorganic chemistry journal instead.

High-Throughput Screening for Discovery of Benchtop Separations Systems for Selected Rare Earth Elements

The most recent report from Schelter and crew is a welcome addition to the body of literature considering solid-liquid separations relevant to intralanthanide separations. The tripodal TriNOx ligand that provided the inspiration for this work was fraught with limitations that would have limited practical implementation for industrial lanthanide separations. The H₃tren-1,2,3-HOPO·TFA ligand presented here is appropriate for publication in Nature Comm due to the significant leap forward in the use of tripodal complexants for lanthanide separations. Tripodal complexants are important to consider seriously for lanthanide separations because their restricted cavity most-likely provides the selective recognition for the small differences in lanthanides across the series.

While we heartily encourage publication, we would encourage the authors to soften some of their statements regarding comparison to current state-of-the-art in the solvent extraction literature. Some of the comparisons made do not seem to be accurate to the best of my understanding, nor are they apples to apples comparisons. While solvent extraction does certainly have its limitations as a “green” process, one of the reasons so much organic solvent is produced is because of the significant throughput possible with solvent extraction at the engineered scale relative to batch solid-liquid separations. Furthermore, it’s not obvious to me that this ligand must stay aqueous soluble. Some aliphatic tails could be placed on the HOPO groups and now one has an extractant that would be possible of high throughput separations and would probably provide comparable separations.

Specific comments for the manuscript are provided below:

- 1) The general statement that “Countercurrent solvent extraction nevertheless requires large volumes of solvents and suffers from low to moderate selectivity for individual rare earths over a single extraction and stripping step” (Paragraph 2) is a bit misleading. I agree that large volumes of solvent are used, but I’ll reiterate that this has to do with the volume of material that can be processed using solvent extraction. Additionally, classifying solvent extraction as having “low to moderate” selective as an inherent part of the process is also incorrect. The separation factors for HDEHP and HEHEHP are comparable to what is presented in this manuscript. The authors suggest otherwise based on a Cyanex product sheet (reference 46), but the peer reviewed literature consistently suggests that the Nd/Dy separation factors are more comparable to what is presented in this document (I’ll touch on this more below).

In short, the authors consistently indicate solvent extraction is not selective, produces large volumes of solvent and (eventually) that the process is slow. This reviewer is more agnostic about how selectivity persists. You can take a ligand and make it precipitate or you can make it organic soluble and an extractant. The selectivity would probably be comparable. Therefore, I encourage them to remove this statement from the document.

- 2) Following in line with this, later in Paragraph 2, the authors comment that “selective precipitation of individual REs from mixtures, isolated with a simple filtration step, could **reduce resource consumption and obviate the need for a large scale countercurrent separations circuit.**” The driver for large-scale countercurrent separations equipment is the volume of material that is being dealt with. The other possible argument for large-scale separations equipment would be if the separation factors were low (as the authors attest) but the literature does not suggest this to be the case. Therefore, we encourage the authors to remove this statement.
- 3) In page 2, paragraph 2, the authors indicate they “noticed a marked dependence on the solution pH for the precipitations of the individual complexes”. What is the equilibrium governing this? Is it the pka of the ligand? Some speculation on this would be appreciated and probably helpful in figuring out how to optimize separations without using HTE methods.
- 4) The authors comment on Page 3, Column 2, paragraph 2 (the authors are encouraged to include line numbers in future submitted documents) that, “Unsurprisingly, the amount of RE precipitated decreased with increasing HCl concentration from 0.10 M to 2.00 M.” It is not obvious to me why this isn’t a surprise. Could the authors elaborate on how they have this intuition in the document? The lanthanides should be well away from hydrolysis at 0.1 M H⁺ (the lowest pH considered). This is somewhat of a follow-up on Comment 3.
- 5) Could the authors comment on mass balances for the studies characterized in Figure 2? That would be much appreciated. It would be good for them to assess this in general for future studies.
- 6) Can the authors comment on why HCl was picked for their aqueous working medium? HCl can pit/chew through the stainless steel used in chemical separations processes significantly. For this reason, HNO₃ is frequently used. I don’t anticipate the results would be significantly different (lanthanides behave in a very similar fashion in either HCl or HNO₃ medium), but some comment from the author would be appreciated.
- 7) The authors note that their best separation occurred after a 24-hour contact. This is one place where single-stage solvent extraction systems perform significantly better than this system. The HDEHP/HEHEHP systems achieve equilibrium in less than 15 minutes (usually around 5 minutes). The faster kinetics is why multi-stage solvent extraction using mixer-settlers or centrifugal contactors is possible. The authors comment that “hundreds of hours are necessary for a solvent extraction system to provide purified rare earths” (Page 4, Column 2, paragraph 1). This is because of the kilograms to tonnes of material being processed in a multi-stage solvent extraction system. Again, we encourage the authors to compare more analogous settings. In a single stage solvent

extraction system, minutes are required to provide comparable separation factors. References indicating this are provided below.

Philip Horwitz, E., McAlister, D.R. and Dietz, M.L., 2006. Extraction chromatography versus solvent extraction: how similar are they?. Separation science and technology, 41(10), pp.2163-2182.

This paper indicates 10 minute contact times were appropriate for the solvent extraction system, extraction chromatographic system using similar ligands suggests 5-10 minutes were necessary for equilibrium with HDEHP or HEH[EHP]. Based on recovering values from graphs, Nd/Dy SFs are ~100 for HDEHP and 310 for HEH[EHP].

Further interesting is the report shows the separation factors are largely dependent on the ligand and not the mode of separation (solvent extraction vs extraction chromatography).

Peppard, D.F., Mason, G.W., Maier, J.L. and Driscoll, W.J., 1957. Fractional extraction of the lanthanides as their di-alkyl orthophosphates. Journal of Inorganic and Nuclear Chemistry, 4(5-6), pp.334-343.

This Peppard paper indicates the Nd/Dy separation factor might be closer to 1000. Separations were demonstrated between 30 second and 3 minute time frames.

Zalupski, P., Nash, K.L. Two-Phase Calorimetry. I. Studies on the Thermodynamics of Lanthanide Extraction by Bis(2-EthylHexyl) Phosphoric Acid. Solvent Extraction & Ion Exchange, 2008, 26(5), 514-533.

This paper indicates equilibrium was achieved in well less than 1000 seconds (probably closer to 250 seconds). Used extraction constants derived elsewhere.

This is a short survey of a significant body of work completed on HDEHP and HEH[EHP]. The authors are encouraged to consider this literature as well as other efforts that have been considered by Peppard, Horwitz, Nash, Kolarik and others that have studied these extractants extensively. In general, the La/Nd separation factor reported by Schelter is probably consistently better than what is observed than the HDEHP/HEH[EHP] class of reagents, but the degree to which it is better varies and in some instances is probably comparable.

- 8) The authors indicate the separation factor between Nd/Dy is ~50 based on a Cyanex product sheet. Much of the literature would approximate these separation factors as higher than provided here. For HDEHP, the Nd/Dy separation factor is around 177. Other reports for HEH[EHP] have separation factors closer to 377. The separation factors for a given solvent extraction system can vary slightly depending on the diluent. I would encourage the authors to use peer-reviewed literature for their comparisons. Some more suggestions are provided below.

Nilsson, M. and Nash, K.L., 2007. A review of the development and operational characteristics of the TALSPEAK process. Solvent extraction and ion exchange, 25(6), pp.665-701.

Kubota, F., Goto, M. and Nakashio, F., 1993. Extraction of rare earth metals with 2-ethylhexyl phosphonic acid mono-2-ethylhexyl ester in the presence of diethylenetriaminepentaacetic acid in aqueous phase. Solvent extraction and ion exchange, 11(3), pp.437-453.

With this in mind, the authors would probably be more appropriate in stating that their separation factors are comparable to the current solvent-extraction state of the art. Even though these ligands are comparable, considering the novelty of the ligand system considered, this manuscript should still be published in Nature Comm. There is much more opportunity for optimizing tripodal ligands for intralanthanide separations than currently exists for the classically used organophosphorus reagents.

- 9) Regarding the ligand stripping studies, I would pause classify the use of 12 M HCl (concentrated HCl) as “mild”. Perhaps the authors should not indicate so in the last sentence of this paragraph.

Reviewer #1 (Remarks to the Author):

- **Comment:** The manuscript entitled as “High-Throughput Screening for Discovery of Benchtop Separations Systems for Selected Rare Earth Elements” is demonstrating that the new ligand $H_31 \cdot TFA$ system, stemming from two reported ligands with their respective high affinity or separation selectivity towards REs, shows high separation factors over a single step for pairs of REs ($SF_{Nd/Dy}=213 \pm 34$ and $SF_{La/Nd}=16.2 \pm 0.2$). Inspired by the observation of different precipitation behavior of 1-RE complexes under the same pH condition, the effects of acid concentration and equivalents of $H_31 \cdot TFA$ on the precipitation of corresponding RE complexes were further studied in a systematic way, setting up an efficient screening method for predicting the optimal separation condition for binary mixtures of rare earth. It was verified by the subsequent detailed experiments, also taking consideration of reaction time and metal concentration. Compared with the current industrial means for the separation of the identical pair of REs by using two commercial extractants, this method herein introduced features the less time- and (harmful) solvent- consumption, higher purity of separated REs and recyclability of the materials. Although the reason for the selectivity of this system is still indefinite, this work indeed showcases a promising separation application from a practical standpoint. Overall, the manuscript is suitable for the Communications Chemistry and I recommend acceptance after minor changes.
 - **Response:** We thank the reviewer for their kind assessment of our work.
- **Comment:** Ligand H_31 was described to exist as the trifluoroacetic acid salt ($H_31 \cdot TFA$), mainly deduced from the obtained crystal structure of it. The crystallization detail of $H_31 \cdot TFA$ should be reported in ESI. Besides, different crystallization conditions can affect the final crystal structure of product. Is it possible that the obtained product co-crystallizes with TFA, thus yielding the trifluoroacetic acid salt?
 - **Response:** X-ray quality crystals were grown by slow evaporation of a water solution of the $H_31 \cdot TFA$ salt. We have added text in the synthetic details and characterization section of the supporting information to clarify this point. It is worth noting that evaporation of a solution of the $H_31 \cdot TFA$ salt in ~ 6 M HCl yielded only crystals of $H_31 \cdot TFA$. During the work up, the final ligand is precipitated from water using TFA, but is isolated as a powder and not as crystalline material. TFA is still present in this powder after drying under reduced pressure.
- **Comment:** How to determine the hydrogen atom on the protonated bridging N atom? Is there reasonable electron density to confirm the existence of this hydrogen atom?
 - **Response:** There was reasonable electron density in the crystal structure close to the tertiary amine that was assigned as the proton from trifluoroacetic acid.
- **Comment:** The direct evidence for the pure form of $H_31 \cdot TFA$ should be the NMR spectroscopy with the internal standard, as the author mentioned. The corresponding spectra should be added in the ESI.
 - **Response:** We have added the spectra to the supporting information (Supplementary Figures 8-9).

- **Comment:** Maybe element analysis will be helpful for further confirmation of the exact form.
 - **Response:** The best fit for the elemental analysis of one batch of ligand contained 2 equivalents of TFA per ligand. We have included these data in the supporting information. However, based on the difference between the crystal structure, and the varying equivalents of TFA obtained from the NMR spectra, we are reluctant to assign the structure as $H_3L \cdot 2TFA$.

- **Comment:** As described in the header of Table 1, there are many factors corresponding to different properties of RE1 and(or) RE2. Among them, the $SF_{RE1/RE2}$ should be corrected as $SF_{RE2/RE1}$, then it can match the mentioned $SF_{Nd/Dy}$ in the main text.
 - **Response:** We have made the recommended correction so the ordering of RE2/RE1 is consistent throughout the main text and the supporting information.

- **Comment:** Two IR spectra are added at the end of ESI. There is no information/explanation about these two spectra in the main text and ESI. Adding one sentence in the figure caption to interpret may be better for the reader to understand.
 - **Response:** These spectra were included to help demonstrate what was present in the solid and filtrate portions of the Nd/Dy separations. We have added a sentence to each of these captions to clarify this point, and added the following text to the discussion section of the manuscript to direct readers to these spectra: “Preliminary results suggest that the filtrate from the optimized Nd/Dy separations mixture comprised a more complicated speciation than simple chloride salts; evident in the infrared spectra of the solid and filtrate portions obtained from the separations experiments (Supplementary Figures 14-15).”

Reviewer #2 (Remarks to the Author):

- **Comment:** The most recent report from Schelter and crew is a welcome addition to the body of literature considering solid-liquid separations relevant to intralanthanide separations. The tripodal TriNO_x ligand that provided the inspiration for this work was fraught with limitations that would have limited practical implementation for industrial lanthanide separations. The H₃tren-1,2,3- HOPO·TFA ligand presented here is appropriate for publication in Nature Comm due to the significant leap forward in the use of tripodal complexants for lanthanide separations. Tripodal complexants are important to consider seriously for lanthanide separations because their restricted cavity most-likely provides the selective recognition for the small differences in lanthanides across the series.
 - **Response:** We thank the reviewer for their kind assessment of our work.

- **Comment:** While we heartily encourage publication, we would encourage the authors to soften some of their statements regarding comparison to current state-of-the-art in the solvent extraction literature. Some of the comparisons made do not seem to be accurate to the best of my understanding, nor are they apples to apples comparisons. While solvent extraction does certainly have its limitations as a “green” process, one of the reasons so

much organic solvent is produced is because of the significant throughput possible with solvent extraction at the engineered scale relative to batch solid-liquid separations.

- **Response:** We thank the reviewer for their thoughtful and thorough review of our work. We have done our best to address the specific comments listed below.
- Furthermore, it's not obvious to me that this ligand must stay aqueous soluble. Some aliphatic tails could be placed on the HOPO groups and now one has an extractant that would be possible of high throughput separations and would probably provide comparable separations.
 - **Response:** This idea is interesting, potentially compatible with solvent extraction infrastructure, and could warrant future investigation. In general, our primary interest has been to develop alternative technologies to solvent extraction while working in a single, benign solvent. The goal here is not to replace solvent extraction at the point of RE mining, but as an alternative during the development of RE recycling facilities.
- **Comment:** Specific comments for the manuscript are provided below: The general statement that “Countercurrent solvent extraction nevertheless requires large volumes of solvents and suffers from low to moderate selectivity for individual rare earths over a single extraction and stripping step” (Paragraph 2) is a bit misleading. I agree that large volumes of solvent are used, but I'll reiterate that this has to do with the volume of material that can be processed using solvent extraction.
 - **Response:** While we understand that industrial-scale chemistry will always require large solvent volumes, this issue is exacerbated by the relatively dilute operating conditions in use. Working under more concentrated conditions would eliminate a significant quantity of solvent waste. We have clarified this point in the introduction with the following text using references provided by the reviewer: “Countercurrent solvent extraction requires large volumes of solvents due to relatively dilute operating conditions (20 ppm ~ 0.1 M RE).”
- **Comment:** Additionally, classifying solvent extraction as having “low to moderate” selective as an inherent part of the process is also incorrect. The separation factors for HDEHP and HEHEHP are comparable to what is presented in this manuscript. The authors suggest otherwise based on a Cyanex product sheet (reference 46), but the peer reviewed literature consistently suggests that the Nd/Dy separation factors are more comparable to what is presented in this document (I'll touch on this more below). In short, the authors consistently indicate solvent extraction is not selective, produces large volumes of solvent and (eventually) that the process is slow. This reviewer is more agnostic about how selectivity persists. You can take a ligand and make it precipitate or you can make it organic soluble and an extractant. The selectivity would probably be comparable. Therefore, I encourage them to remove this statement from the document.
 - **Response:** We have restructured this paragraph of the introduction to provide a more balanced assessment of countercurrent solvent extraction and included references recommended by the reviewer. The introduction now reads: “RE separations are performed industrially using countercurrent solvent extraction. Countercurrent solvent extraction requires large volumes of solvents due to

relatively dilute operating conditions (20 ppm ~ 0.1 M RE).¹⁰⁻¹² Opportunities remain to improve selectivity for individual rare earths over a single extraction and stripping step. Toward these goals, researchers have developed novel ligands,¹³ ionic liquids,^{14,15} and extractants.^{16-18,}

- **Comment:** Following in line with this, later in Paragraph 2, the authors comment that “selective precipitation of individual REs from mixtures, isolated with a simple filtration step, could **reduce resource consumption and obviate the need for a large scale countercurrent separations circuit.**” The driver for large-scale countercurrent separations equipment is the volume of material that is being dealt with. The other possible argument for large-scale separations equipment would be if the separation factors were low (as the authors attest) but the literature does not suggest this to be the case. Therefore, we encourage the authors to remove this statement.
 - **Response:** We have removed mention of solvent extraction from the above referenced statement.
- **Comment:** In page 2, paragraph 2, the authors indicate they “noticed a marked dependence on the solution pH for the precipitations of the individual complexes”. What is the equilibrium governing this? Is it the pK_a of the ligand? Some speculation on this would be appreciated and probably helpful in figuring out how to optimize separations without using HTE methods.
 - **Response:** We believe the difference in precipitation among the REs as a function of pH is likely due to differences in the pK_a values of the ligand for each RE. We have included the following statement in the discussion section with this hypothesis: “The origin of the selectivity of this system is potentially due to differences in the pK_a values of the ligand among the different REs, which results in the formation of species with differing solubility at varying pHs.”
- **Comment:** The authors comment on Page 3, Column 2, paragraph 2 (the authors are encouraged to include line numbers in future submitted documents) that, “Unsurprisingly, the amount of RE precipitated decreased with increasing HCl concentration from 0.10 M to 2.00 M.” It is not obvious to me why this isn’t a surprise. Could the authors elaborate on how they have this intuition in the document? The lanthanides should be well away from hydrolysis at 0.1 M H^+ (the lowest pH considered). This is somewhat of a follow-up on Comment 3.
 - **Response:** We have removed the word “unsurprisingly” from the manuscript. In aqueous conditions, 0.1 M H^+ is adequate to prevent acid hydrolysis of simple RE salts to form hydroxide/oxide species. However, in our experience, RE complexes will generally hydrolyze under acidic conditions to form the RE salt and return the protonated ligand. For this reason, we were not surprised that our yields would decrease with increasing concentrations of acid.
- **Comment:** Could the authors comment on mass balances for the studies characterized in Figure 2? That would be much appreciated. It would be good for them to assess this in general for future studies.

- **Response:** The yields from the HTE study were not determined by measuring the mass of precipitate generated, but calculated based on differences in the RE concentration in the filtrate between experimental wells and positive control wells. With this method, we do not have to thoroughly dry the solid portion, or account for excess ligand to obtain yields. We have added a sentence to the Methods section of the manuscript to refer readers to the supporting information for this calculation, and clarified the rationale behind this methodology in the supporting information.
- **Comment:** Can the authors comment on why HCl was picked for their aqueous working medium? HCl can pit/chew through the stainless steel used in chemical separations processes significantly. For this reason, HNO₃ is frequently used. I don't anticipate the results would be significantly different (lanthanides behave in a very similar fashion in either HCl or HNO₃ medium), but some comment from the author would be appreciated.
 - **Response:** During preliminary investigations into separations we tested both HCl and HNO₃ as the aqueous working medium and found the separations performances to be equivalent within error. We chose to continue all studies in HCl due to the current supply of RECl₃ salts in our lab. We appreciate the reviewer's point and will make the suggested change in future studies.
- **Comment:** The authors note that their best separation occurred after a 24-hour contact. This is one place where single-stage solvent extraction systems perform significantly better than this system. The HDEHP/HEHEHP systems achieve equilibrium in less than 15 minutes (usually around 5 minutes). The faster kinetics is why multi-stage solvent extraction using mixer-settlers or centrifugal contactors is possible. The authors comment that "hundreds of hours are necessary for a solvent extraction system to provide purified rare earths" (Page 4, Column 2, paragraph 1). This is because of the kilograms to tonnes of material being processed in a multi-stage solvent extraction system. Again, we encourage the authors to compare more analogous settings. In a single stage solvent extraction system, minutes are required to provide comparable separation factors. References indicating this are provided below.
 - **Response:** We thank the reviewer for the referenced literature detailing solvent extraction systems. We have removed direct comparisons of time requirements for a single extraction stage between our system and those performed in solvent extraction in agreement with the reviewer's suggestions.
 - Philip Horwitz, E., McAlister, D.R. and Dietz, M.L., 2006. Extraction chromatography versus solvent extraction: how similar are they? Separation science and technology, 41(10), pp.2163-2182. This paper indicates 10 minute contact times were appropriate for the solvent extraction system, extraction chromatographic system using similar ligands suggests 5-10 minutes were necessary for equilibrium with HDEHP or HEH[HEP]. Based on recovering values from graphs, Nd/Dy SFs are ~100 for HDEHP and 310 for HEH[EHP].
 - Further interesting is the report shows the separation factors are largely dependent on the ligand and not the mode of separation (solvent extraction vs extraction chromatography). Peppard, D.F., Mason, G.W., Maier, J.L. and Driscoll, W.J., 1957. Fractional extraction of the lanthanides as their di-alkyl orthophosphates.

Journal of Inorganic and Nuclear Chemistry, 4(5-6), pp.334-343. This Peppard paper indicates the Nd/Dy separation factor might be closer to 1000. Separations were demonstrated between 30 second and 3 minute time frames.

- Zalupski, P., Nash, K.L. Two-Phase Calorimetry. I. Studies on the Thermodynamics of Lanthanide Extraction by Bis(2-Ethylhexyl) Phosphoric Acid. Solvent Extraction & Ion Exchange, 2008, 26(5), 514-533. This paper indicates equilibrium was achieved in well less than 1000 seconds (probably closer to 250 seconds). Used extraction constants derived elsewhere.
 - This is a short survey of a significant body of work completed on HDEHP and HEH[EHP]. The authors are encouraged to consider this literature as well as other efforts that have been considered by Peppard, Horwitz, Nash, Kolarik and others that have studied these extractants extensively.
 - In general, the La/Nd separation factor reported by Schelter is probably consistently better than what is observed than the HDEHP/HEH[EHP] class of reagents, but the degree to which it is better varies and in some instances is probably comparable.
 - **Response:** We have included several of the references provided above in our manuscript, and would like to again thank the reviewer for providing them.
- **Comment:** The authors indicate the separation factor between Nd/Dy is ~50 based on a Cyanex product sheet. Much of the literature would approximate these separation factors as higher than provided here. For HDEHP, the Nd/Dy separation factor is around 177. Other reports for HEH[EHP] have separation factors closer to 377. The separation factors for a given solvent extraction system can vary slightly depending on the diluent. I would encourage the authors to use peer-reviewed literature for their comparisons. Some more suggestions are provided below.
 - Nilsson, M. and Nash, K.L., 2007. A review of the development and operational characteristics of the TALSPEAK process. Solvent extraction and ion exchange, 25(6), pp.665-701.
 - Kubota, F., Goto, M. and Nakashio, F., 1993. Extraction of rare earth metals with 2-ethylhexyl phosphonic acid mono-2-ethylhexyl ester in the presence of diethylenetriaminepentaacetic acid in aqueous phase. Solvent extraction and ion exchange, 11(3), pp.437-453.
 - With this in mind, the authors would probably be more appropriate in stating that their separation factors are comparable to the current solvent-extraction state of the art.
 - **Response:** We thank the reviewer for providing these references. We have included one set of approximate separation factors that are comparable to those we were able to achieve, obtained from one of the recommended references, in Table 2. We have included the following text in the Separations Experiments Based on HTE section of the manuscript as well: “However, it is worth noting that performance of phosphorous-based extractants is highly dependent on the diluents used, with some reports achieving comparable separations to H₃I·TFA.” We have added the following text to the Discussion section as well: “Our system was found to

be comparable to and in some cases outperform currently relevant industrial countercurrent solvent extractants HEHEHP..."

- Even though these ligands are comparable, considering the novelty of the ligand system considered, this manuscript should still be published in Nature Comm. There is much more opportunity for optimizing tripodal ligands for intralanthanide separations than currently exists for the classically used organophosphorus reagents.
 - **Response:** We wholeheartedly agree with the reviewer's recommendation.
- **Comment:** Regarding the ligand stripping studies, I would pause classify the use of 12 M HCl (concentrated HCl) as "mild". Perhaps the authors should not indicate so in the last sentence of this paragraph.
 - **Response:** We have removed the sentence fragment "under relatively mild conditions" in agreement with the reviewer's comment.

Reviewer #3 (Remarks to the Author):

- **Comment:** The manuscript contains some interesting experimental findings on the selective precipitation of Ln (La, Nd and Dy) with a newly synthesized ligand. However, the paper aims at developing a technology for Ln recycling, but the proposed approach is far from being superior to the current technologies.
 - **Response:** Over the course of addressing Reviewer #2's comments, we have softened the comparison of our system to current technologies (separation factors achieved in a single step, time requirements, solvent usage). However, we contend that our system offers the advantage of being in purely aqueous media to accomplish separations, where selectivity can be altered using adjustments to the primary coordination sphere.
- **Comment:** First- the authors claim that "... solvent extraction can require hundreds of hours to achieve purified REs, compared to the less than 48 hours required for our system including ligand recovery steps (vide infra).16" Reference 16 is hardly relevant here as it describes the application of Aliquat 336 in a certain system. While the authors compare their method with HEHEHP and Cyanex 572 in terms of separation factors, they do not cite any literature describing the corresponding processes (Ref 46 does not point to any process conditions). As these processes are counter-current, very high overall separation factors (1,000-10,000) can be accomplished within 10-20 stages, and the duration of the process should not exceed several hours even if the slowest equipment (e.g. mixer-settler) is used.
 - **Response:** While addressing Reviewer #2's comments, we have removed direct comparisons of time requirements between our system and those performed in solvent extraction. The above-mentioned reference detailing Aliquat 336 does include reference to HEHEHP, but has ultimately been removed from the manuscript.
- **Comment:** Second- The SF for La/Nd for the proposed method is only 16, and it would take 3-4 precipitation steps (6-8 days) to get to a decent overall separation.

- **Response:** We agree with the reviewer that multiple iterations to have significant purification of a La/Nd mixture. We have clarified the statement quoted above to state “purified Nd and Dy” instead of “purified REs” to avoid implying our process performs as well for La/Nd mixtures as it does for Nd/Dy.
- **Comment:** Third issue is the Ln recovery and the recycling of the chemicals: according to the paper: " 1·Dy could be dissolved in 0.4 mL 12 M HCl, presumably forming $H_31 \cdot HCl$ and $DyCl_3 \cdot nH_2O$ in solution. Addition of 2.0 mL EtOH resulted in the formation of a precipitate, which was determined to be H31 with a residual ~12% 1·Dy by 1H NMR analysis (see ESI for experimental details). Considering the ligand could be reused for additional Nd/Dy separations, this minor impurity does not pose an operational issue. This stripping step was able to recover 84% of the ligand and 86% $DyCl_3$ under relatively mild conditions while using minimal solvent volumes." It is unclear why using concentrated HCl is called "mild conditions".
 - **Response:** We have removed the sentence fragment “under relatively mild conditions” in agreement with the reviewer’s comment.
- **Comment:** Using ethanol to precipitate the ligand, following by the filtration of the solid and its dissolution will further increase processing time.
 - **Response:** The increase in processing time required for ligand recovery was included in the manuscript in its original form, but the sentence was removed to avoid direct comparison of time requirements between our work and solvent extraction.
- **Comment:** Also, returning 12% Dy to the separation step will eventually saturate the system with Dy and will lower the purity of Nd.
 - **Response:** RE recycling mixtures generally have ~5% Dy/Nd, so we contend this incremental increase in Dy will not significantly affect the system. Furthermore, Dy is the more valuable of the two elements, and we are more concerned with recovering pure Dy than Nd. For these reasons, we are not concerned with returning the additional Dy to the system.
- **Comment:** The manuscript should not be published in this current form; it should be submitted to a specialized inorganic chemistry journal instead.
 - **Response:** We respectfully disagree with the reviewer about publishing this manuscript here in a revised form, but would like to thank the reviewer for their comments and the opportunity to improve our manuscript.

REVIEWERS' COMMENTS:

Reviewer #2 (Remarks to the Author):

We thank the authors for their thoughtful responses to the reviewers comments. My concerns have been addressed and I recommend publication.

REVIEWERS' COMMENTS:

Reviewer #2 (Remarks to the Author):

We thank the authors for their thoughtful responses to the reviewers comments. My concerns have been addressed and I recommend publication.